# Predicting fall risk using multiple mechanics-based metrics for a planar biped model

Daniel Williams[ID]¤, Anne E. Martin[ID]*

Department of Mechanical Engineering, The Pennsylvania State University, University Park, PA, United States of America

¤ Current address: Department of Mechanical Engineering, University of West Florida, Pensacola, FL, United States of America

* aem34@psu.edu

**Data Availability Statement:** All relevant data are within the manuscript and its Supporting information files.

**Funding:** AEM received a grant from the National Science Foundation (award 1727540) for this work.

## Abstract

For both humans and robots, falls are undesirable, motivating the development of fall prediction models. Many mechanics-based fall risk metrics have been proposed and validated to varying degrees, including the extrapolated center of mass, the foot rotation index, Lyapunov exponents, joint and spatiotemporal variability, and mean spatiotemporal parameters. To obtain a best-case estimate of how well these metrics can predict fall risk both individually and in combination, this work used a planar six-link hip-knee-ankle biped model with curved feet walking at speeds ranging from 0.8 m/s to 1.2 m/s. The true number of steps to fall was determined using the mean first passage times from a Markov chain describing the gaits. In addition, each metric was estimated using the Markov chain of the gait. Because calculating the fall risk metrics from the Markov chain had not been done before, the results were validated using brute force simulations. Except for the short-term Lyapunov exponents, the Markov chains could accurately calculate the metrics. Using the Markov chain data, quadratic fall prediction models were created and evaluated. The models were further evaluated using differing length brute force simulations. None of the 49 tested fall risk metrics could accurately predict the number of steps to fall by themselves. However, when all the fall risk metrics except the Lyapunov exponents were combined into a single model, the accuracy increased substantially. These results suggest that multiple fall risk metrics must be combined to obtain a useful measure of stability. As expected, as the number of steps used to calculate the fall risk metrics increased, the accuracy and precision increased. This led to a corresponding increase in the accuracy and precision of the combined fall risk model. 300 step simulations seemed to provide the best tradeoff between accuracy and using as few steps as possible.

## Introduction

Falling can be an expensive and debilitating issue both for elderly adults [1, 2] and robots, and predicting fall risk could help mitigate these issues. Ideally, increased fall risk would be detected a long time before a fall happens to allow preventative measures to be taken.

The funders had no role in study design, data collection and analysis, decision to publish, or preparation of the manuscript.

**Competing interests:** The authors have declared that no competing interests exist.

Unfortunately, despite a tremendous amount of work, accurately predicting falls is still an open question. There have been many fall risk factors proposed, with varying degrees and types of validation performed. Some of these risk factors focus on subject-related factors [3], some examine level of physical activity or ability [3, 4], and some examine the biomechanics of the subjects' gaits [5–9]. Even when just considering the biomechanics, many potential stability metrics exist, each of which identifies a single mode of instability and likely cannot individually quantify the risk of falling. Instead, multiple metrics can be combined into a single fall risk model which likely increases the predictive ability compared to single metrics. Because most studies have only considered a single metric in isolation, it is difficult to tell the relative accuracy of each metric, and it is unknown how the accuracy improves when the metrics are combined.

One class of (bio)mechanical fall risk metrics are based on ground reaction forces (GRF), including the extrapolated center of mass (XCoM) [10], the zero moment point (ZMP) [11], and the foot rotation index (FRI) [12]. The XCoM is found by adding a velocity term to the ground projection of the center of mass [10]. Generally, stability is assumed to decrease the farther the XCoM extends beyond the basin of support. This has been supported in correlational human walking studies [13, 14]. The ZMP or center of pressure is the point on the ground where the GRF has no net moment [11, 15]. The ZMP stability criterion dictates that the ZMP must stay within the basin of support for a biped gait to be stable. An extension of this is the FRI, which is the point on the ground where the net GRF would have to act to keep the foot from rotating [12]. Although the FRI point may extend beyond the basin of support during a stable gait, a common interpretation is that the further away the FRI point is from the basin of support, the less stable the biped becomes. Although the ZMP has been used to detect imminent falls [16], there has been little research done to relate ZMP or FRI with predicted fall risk.

Stability can also be assessed using analysis methods from dynamical systems, including Floquet multipliers [17] and Lyapunov exponents [18]. The first of these, Floquet multipliers, essentially measures how quickly a system returns to its periodic trajectory after a perturbation [17, 19]. However, the assumption of exact periodicity is generally not valid for bipedal gait, so Floquet multipliers may not be an appropriate measure [19, 20]. Lyapunov exponents expand on Floquet multipliers by dropping the assumption of periodicity [18]. They quantify the average exponential rate of divergence between two neighboring trajectories. This divergence quantifies how sensitive the gait is to small perturbations, and a higher value indicates that the gait is less stable. Both short-term Lyapunov exponents calculated between 0–1 strides and long-term Lyapunov exponents calculated from 4–10 strides have been used to evaluate the stability of a bipedal system. However, the length of the sample size affects the results [21]. There has been some evidence showing the correlation of Lyapunov exponents [7, 22] and maximum Floquet multipliers [23] with fall risk in retrospective human studies.

Other metrics, including variability [7] and mean spatiotemporal (step duration, step length, and speed) measures [6], have also been used to assess fall risk. Increased gait variability is thought to increase fall risk by increasing the chance that the system will end up in a state it cannot recover from [7, 24, 25]. Variability can be measured for spatiotemporal data or for continuous data, such as joint angles and joint velocities. Both experimental and simulation studies have shown a correlation between increased variability and decreased stability [7, 25–27], though this correlation was not always linear [24]. However, studies have also shown the converse [26, 28]. In addition, healthy, stable human gait contains variability [29], so the presence of variability does not necessarily indicate instability [30]. Spatiotemporal measures, particularly decreased walking speed, are sometimes [6, 31], but not always [32], correlated with fall risk in humans. Therefore, these metrics may not indicate fall risk by themselves.

A major challenge in assessing these potential fall risk metrics is collecting the needed data. Even fall prone humans only fall once or twice a year [2], so human subject studies require either self-reported fall history or long-term monitoring to record falls. An alternative is to use dynamic simulations of walking to generate highly controlled data quickly [24, 30, 33]. While these models do not contain the full complexity of human gait, they provide far more control of the experiment and allow researchers to obtain a best-case estimate of how accurate a fall prediction model is. In addition, simulations can generate steady walking for an arbitrary number of steps, making it easier to compute some potential fall risk metrics that require sequences of hundreds of steps. Thus, simulations can also be used to explore the trade-offs between using many steps to compute a metric accurately and using fewer steps to make a human experimental protocol more feasible.

Using dynamic simulations and quadratic regressions, this paper determined the best case fall prediction accuracy for a total of 49 biomechanical-based fall risk metrics both individually and combined into a single model. In addition, the improvement in accuracy as the number of steps increased was determined. To generate the needed data, this work used a planar six-link hip-knee-ankle biped model with curved feet walking at speeds ranging from 0.8 m/s to 1.2 m/s. Both the true number of steps to fall [34] and the proposed metrics were determined using Markov chains describing the gaits. A secondary contribution of this paper was the development and validation of the metric calculations using the Markov chain. The models were validated using brute force simulations of both the training gaits as well as new validation gaits.

## Materials and methods

### Biped models

This work used the biped models and gaits from [35]. A brief overview is given here for completeness and the reader is referred to [35] for details. The model itself was a planar six-link biped with curved feet, knee and ankle joints, and a point mass at the hip (leg length 1.16 m, total mass 86.7 kg, Fig 1). The biped configuration was described using eight coordinates: $q_1$ described the absolute orientation, $q_2 \ldots q_6$ described the joint angles, and $q_7$ and $q_8$ described the absolute planar position of the hip. The hip, knee, and ankle joints were actuated using ideal actuators, while the stance foot was assumed to roll along the ground without slip. Because the biped was underactuated, it could fall if the actuated joints did not move appropriately. In addition, a step could fail by requiring a coefficient of friction greater than 0.7 and/or requiring a negative vertical ground reaction force. This work treated all three failure modes the same.

Each step consisted of a finite-time single support period and a double support period during which the stance leg switched. The single support period was controlled using feedback linearization. Two models for the double support period were used—a finite time double support period that was controlled using feedback linearization and an instantaneous double support period that had no control. To allow the finite-time double support period, the ground was slightly complaint.

A total of ten gaits from [35] were used to evaluate each fall risk metric individually as well as develop and evaluate the fall risk model (Table 1). Five gaits had a finite-time double support period and five gaits had an instantaneous double support period. The periodic gait speeds ranged from 0.8 to 1.2 m/s in 0.1 m/s increments. Four additional gaits were created to further validate the fall risk model (Table 1). In all cases, the gaits minimized the specific energetic cost of transport at a given speed [36]. As described in the next subsection, each gait was perturbed every step. There were a total of three perturbation timings, leading to a total of 30 training conditions (5 speeds × 2 double support models × 3 perturbation timing conditions)

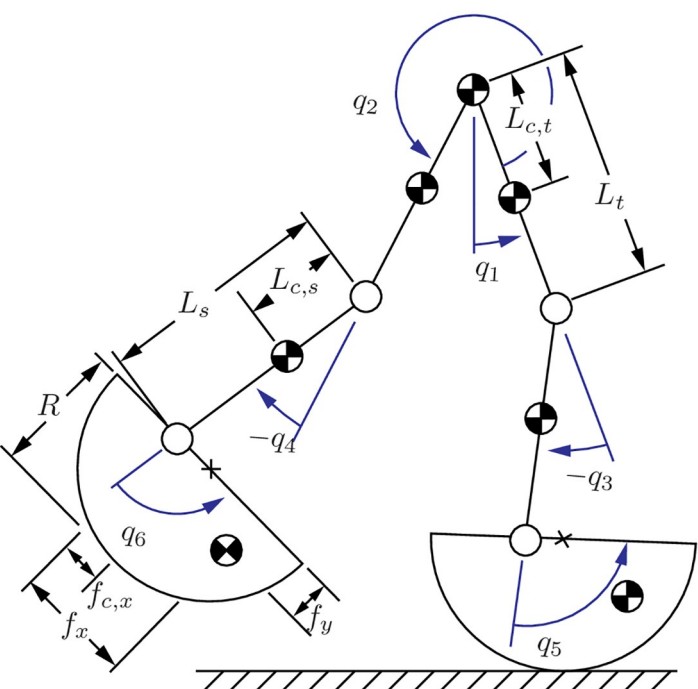

**Fig 1. Six link biped model.** The unactuated angle is $q_1$ and the actuated angles are $q_2 \ldots q_6$.

and 12 validation conditions (2 speeds × 2 double support models × 3 perturbation timing conditions).

## Gait data

The gaits used were metastable, meaning they will eventually, but not quickly, fail when experiencing small, random perturbations. Because of this, they were analyzed as discrete-time

**Table 1. Details for gaits used in this paper.** The first column indicates if the gait was used to develop the fall prediction model ("Train") or validate it ("Val"). The second column indicates the double support type, either instantaneous ("IS") or finite-time ("FT"). The next three columns give the speed, step length, and step duration for the periodic gait. The final three columns give the mean first passage time when the perturbations were applied at the beginning of double support ("$DS_0$"), at the beginning of single support ("$SS_0$"), or halfway through single support ("$SS_{0.5}$").

| Use | DS Type | $v$ (m/s) | $L$ (m) | $T$ (s) | MFPT | | |
|---|---|---|---|---|---|---|---|
| | | | | | $DS_0$ | $SS_0$ | $SS_{0.5}$ |
| Train | IS | 0.80 | 1.05 | 1.32 | 185.1 | 185.1 | $2.840 \times 10^7$ |
| Train | IS | 0.91 | 1.04 | 1.15 | 154.3 | 153.0 | $8.990 \times 10^{10}$ |
| Train | IS | 1.00 | 1.04 | 1.04 | 472.7 | 462.5 | $4.503 \times 10^{11}$ |
| Train | IS | 1.10 | 1.04 | 0.95 | $6.587 \times 10^6$ | $6.587 \times 10^6$ | $1.877 \times 10^{14}$ |
| Train | IS | 1.20 | 1.02 | 0.85 | $6.367 \times 10^4$ | $6.525 \times 10^4$ | $2.252 \times 10^{14}$ |
| Train | FT | 0.79 | 0.99 | 1.25 | $1.988 \times 10^3$ | 449.0 | $1.501 \times 10^{15}$ |
| Train | FT | 0.88 | 0.97 | 1.11 | $3.655 \times 10^3$ | $2.015 \times 10^3$ | $1.126 \times 10^{15}$ |
| Train | FT | 0.98 | 0.97 | 1.00 | $5.275 \times 10^3$ | $2.859 \times 10^6$ | $6.434 \times 10^{14}$ |
| Train | FT | 1.10 | 0.98 | 0.89 | $1.083 \times 10^4$ | 194.1 | $6.397 \times 10^4$ |
| Train | FT | 1.13 | 0.95 | 0.84 | $6.770 \times 10^4$ | 3.847 | 6.499 |
| Val | IS | 0.95 | 0.91 | 0.96 | $1.067 \times 10^3$ | $1.067 \times 10^3$ | $2.501 \times 10^{12}$ |
| Val | IS | 1.05 | 0.89 | 0.85 | $5.337 \times 10^4$ | $6.475 \times 10^4$ | $1.838 \times 10^{14}$ |
| Val | FT | 0.95 | 1.03 | 1.08 | $7.304 \times 10^{10}$ | $6.005 \times 10^{14}$ | $7.506 \times 10^{14}$ |
| Val | FT | 1.05 | 1.00 | 0.95 | $1.279 \times 10^{13}$ | $3.002 \times 10^{15}$ | $9.007 \times 10^{15}$ |

Markov chains to estimate the number of steps to fail [34]. To create the Markov chains, two state-transition meshes were defined for each case. First, a deterministic state-transition mesh was found by systematically simulating the biped for one step for every possible perturbation magnitude and from every possible start-of-step state. In this work, three deterministic state-transition meshes were found for each periodic gait, corresponding to perturbations applied 1) at the start of the double support period, 2) at the start of the single support period, or 3) half-way through the single support period. For gaits with an instantaneous double support period, perturbations 1) and 2) occurred immediately before and after double support, respectively. These perturbations were created by instantaneously changing the velocities of the biped, simulating a horizontal impulsive push on the hip. They ranged in size from −0.200 to + 0.200 m/s with a 0.002 m/s discretization spacing. For each transition in the mesh, the biped states over the entire trajectory were re-sampled to 100 points and saved. For cases when more than one perturbation magnitude resulted in the same initial and final state, the trajectory for the median perturbation was saved. Then, a stochastic state transition mesh and associated Markov chain were found from the deterministic state-transition mesh by assuming the perturbations were normally distributed according to $\mathcal{N}\left(0, \left(\frac{0.01}{3}\right)^2\right)$. The second largest eigenvalue of the stochastic state transition mesh $\lambda_2$ gave the mean first passage time (MFPT)

$$\text{MFPT} \approx \frac{-1}{log(\lambda_2)},\qquad(1)$$

where the MFPT is the expected number of steps before failure. The MFPT from the Markov chains were used as the true number of steps to failure, and are hereafter referred to as the true MFPT. The MFPT varied widely, ranging from $\approx 4$ to $10^{16}$ steps (Table 1). As described later, data from each Markov chain was also used to estimate each fall risk metric, referred to as the Markov chain metric. These data were used to develop the fall prediction models.

Series of brute force gait simulations were also created. To generate these data, an initial set of 10 random perturbation vectors of length 1000 were created. These vectors were then used to simulate each gait with perturbations every step for 1000 steps. After simulation, the data was re-sampled to 100 points per step. The same vectors of perturbations were used for every gait. However, because the gaits sometimes failed prior to the full 1000 steps, additional trials were performed if needed until the resulting simulations could be divided into 200 individual trajectories of 50 steps. These simulations were used to quantify how fall prediction accuracy improved as the number of steps used increased. To do this, the brute force simulations were subdivided into several groups with a smaller number of steps. The number of steps in each group were $N = \{10 \quad 20 \quad 30 \quad 40 \quad 50 \quad 60 \quad 70 \quad 80 \quad 90 \quad 100 \quad 200 \quad 300 \quad 500 \quad 1000\}$. For each $n \in N$, each simulation was first subdivided into segments of $n$ steps. Then all of the fall risk metrics were calculated for every segment of $n$ steps. The fall risk metrics found using the brute force data will be referred to as the brute force metrics.

## Calculating fall risk metrics

To capture as much of the gait behavior as possible in order to accurately predict MPFT, 49 fall risk metrics were tested. Each of these metrics were believed to be related to fall risk, though the exact relationship was not known. Once each of the metrics were calculated, they were centered and normalized so that the mean of the metric became 0 and the standard deviation of the metric became 1. This was done so that metrics with a relatively high mean value would not influence the fall risk model more than those with relatively low mean values, e.g. long-term Lyapunov exponents are typically 2 orders of magnitude smaller than short-term

Lyapunov exponents. The normalization was created using the Markov chain metrics and then applied to the brute force metrics.

**Fall risk metrics.** To quantify how the GRF behaves during perturbed walking, the mean and standard deviation of the extrapolated center of mass (XCoM) and the foot rotation index (FRI) were calculated. The XCoM is given by [10]

$$\text{XCoM} = (x_7 - x_{st}) + \dot{x}_7 \sqrt{\frac{\ell}{g}}, \tag{2}$$

where $\ell = \sqrt{(x_7 - x_{st})^2 + (x_8 - y_{st})^2}$ is the distance between the hip and stance foot contact (or leading foot during double support), $x_7$ and $x_8$ are the horizontal and vertical positions of the hip, $x_{st}$ and $y_{st}$ are the horizontal and vertical positions of the stance foot (or leading foot during double support) contact, and $g$ is gravitational acceleration. The FRI is calculated by determining the point on the ground where the GRF would have to act to prevent the foot from rotating [12]:

$$\text{FRI} = \frac{\overrightarrow{OG} \times \overrightarrow{F_g} - \tau_a - \overrightarrow{OA} \times \overrightarrow{F_a}}{F_n} - x_{st}, \tag{3}$$

where $O$ is the stationary origin, $G$ is the center of mass of the foot, $A$ is the stance ankle joint, $F_g$ is the gravitational force on the foot, $F_n$ is the normal GRF, $F_a$ is the stance ankle force, $\tau_a$ is the stance ankle torque, and $x_{st}$ is the horizontal position of the stance foot. For both XCoM and FRI, the maximum value was found for each step, and then the mean and standard deviation of these maximums over all steps served as the fall risk metrics.

The second set of metrics were the short- and long-term Lyapunov exponents for the position and velocity of each of each joint [37]. To calculate the Lyapunov exponent, the state space was $X(t) = \begin{bmatrix} x(t) & x(t+\tau) & \dots & x(t+4\tau) \end{bmatrix}$ where $X \in \mathbb{R}^{(100 n_{\text{steps}} - 4\tau) \times 5}$ is the reconstructed state space, $n_{\text{steps}}$ is the number of steps used, $\tau = 15$ is the time delay, and $x$ is one trajectory [9, 21, 37]. For each point in the state space, the initially nearest neighbor for each $X(j)$ was determined as the point $X(k)$ such that $j - 50 > k$ or $k > j + 50$ and $\|X(j) - X(k)\| < \|X(j) - X(l)\|$ for all valid $l$, i.e. $X(j)$ and $X(k)$ are from separate steps and the Euclidean distance between $X(j)$ and $X(k)$ is smaller than every other such point in $X$. Once the nearest neighbor for each point was found, the Euclidean distance $d_j(i)$ between all future points was calculated. The mean divergence curve was calculated using

$$y_\lambda(i) = \frac{1}{\Delta t} \langle \ln[d_j(i)] \rangle. \tag{4}$$

where $\langle \cdot \rangle$ is the average over $j$. This mean divergence curve was then used to calculate both the short term Lyapunov exponent $\lambda_S$ from a linear fit to the divergence curve between 0 and 1 strides ($1 \le i \le 200$) and the long term Lyapunov exponent $\lambda_L$ from a linear fit to the divergence curve between 4 and 10 strides ($801 \le i \le 2000$).

The next set of parameters were the mean and standard deviation of speed, step length, and step duration. The final set of parameters were the average variability of each individual biped coordinate, calculated by finding the standard deviation of that coordinate at each time point, taking the log of each standard deviation, and then averaging over the entire step. The horizontal hip position was not included because it was not cyclical since the biped moved forward.

Thus, a total of 49 metrics were considered (two each for XCoM, FRI, walking speed, step length, and step duration, twelve each for short- and long-term Lyapunov exponents, and fifteen joint variability metrics).

**Calculation details when using a Markov chain.** Calculating each metric was straightforward for the brute force data. Calculating the metrics was more challenging for the Markov chain data because the Markov chain is a collection of possible steps the biped could take and the probability of taking each of these possible steps. To calculate gait parameters for the overall gait, gait parameters from the individual steps in the Markov chain need to be processed.

Because the stochastic Markov chains used to model bipedal gait include the failure state, they are absorbing Markov chains [38] and can be written as

$$T^s = \begin{pmatrix} \mathbf{Q} & \mathbf{R} \\ \mathbf{0} & 1 \end{pmatrix},$$

(5)

where $\mathbf{0}$ is a zero vector, $\mathbf{Q} \in \mathbb{R}^{t \times t}$, and $\mathbf{R} \in \mathbb{R}^{1 \times t}$. The first $t$ rows and columns, i.e. states in the Markov chain, are transient and the last state is absorbing (i.e. once reached the biped cannot recover). Therefore, $\mathbf{Q}$ is used to infer the behavior of the biped before it fails. Most fall risk metrics were found by taking a weighted mean and standard deviation over the transitions (steps) in $\mathbf{Q}$. First, the weighted mean and standard deviation of a metric for each step $p_i$ was found via

$$\mu_i = \frac{\sum_{j=1}^{t} P(s_{ij}) \rho_{ij}}{\sum_{j=1}^{t} P(s_{ij})}$$

(6)

$$\sigma_i^2 = \frac{\sum_{j=1}^{t} P(s_{ij}) (\rho_{ij} - \mu_i)^2}{\sum_{j=1}^{t} P(s_{ij})},$$

(7)

where $\rho_{ij}$ is the value of the parameter being analyzed when transitioning from state $p_i$ to state $p_j$, $\mu_i$ is the mean of $\rho_{ij}$ over the step when the biped begins in state $p_i$, $\sigma_i$ is the standard deviation of $\rho_{ij}$ over the step when the biped begins in state $p_i$, and $P(s_{ij})$ is the probability of perturbation $s_{ij}$. Once $\mu_i$ and $\sigma_i$ were found, they were used to find the overall weighted mean and standard deviation. Because the starting state of the biped is the nominal step $p_1$, $n_{1i} \in \mathbf{N} = (\mathbf{I} - \mathbf{Q})^{-1}$ is the expected number of times that the biped will be in state $p_i$. Therefore, $n_{1i}$ were used as frequency weights to calculate the overall weighted mean $\mu$ and standard deviation $\sigma$

$$\mu = \frac{\sum_{i=1}^{t} n_{1i} \mu_i}{\sum_{i=1}^{t} n_{1i}}$$

(8)

$$\sigma^2 = \frac{\sum_{i=1}^{t} n_{1i} (\sigma_i^2 + (\mu_i - \mu)^2)}{\sum_{i=1}^{t} n_{1i}}.$$

(9)

For all parameters except the Lyapunov exponents, Eqs 6–9 replaced the normal mean and standard deviation calculations when finding the fall risk metric from the Markov chain.

Calculating the Lyapunov exponents from the Markov chain was not as easy. Eqs 6–9 cannot be directly used to calculate the Lyapunov exponents because the calculation requires a sequence of at least 20 steps, and pilot testing indicated that at least 50 steps were needed for meaningful results. Calculating the exact Lyapunov exponent using every possible sequence of steps would generate $\approx 10^{17}$ unique trajectories, each with a calculable, but very small, probability of occurring. Calculating the Lyapunov exponents for this number of trajectories is intractable. Instead, the Lyapunov exponents were estimated by starting at the 20 states the biped was most likely to be in, i.e. the states with the largest values of $n_{1i}$. For each starting state, a sequence of 50 steps was generated using 20 random perturbation vectors. This yielded

a total of 400 random trajectories through the Markov chain. For each trajectory, the Lyapunov exponents were calculated as described earlier. Because these trajectories were generated using the Markov chain, there is a chance that the exact same step can appear more than once in a single trajectory. When this happens, the Euclidean distance $d_j(i)$ becomes 0 and Eq 4 is undefined. Therefore, $d_j(i)$ was limited to be no less than the numerical tolerance of the simulation ($1 \times 10^{-7}$). Once the Lyapunov exponents were calculated for all 400 trajectories, the weighted means of these 400 values were calculated to find the overall Lyapunov exponents. The weight for each gait trajectory was the product of the probability of the given starting state ($n_{1i}$) and the probability of the perturbation vector used (0.05 because the vectors were generated randomly).

Once all of the fall risk metrics were calculated from the Markov chains, they were used to create the metric normalization. For each metric, the values for all training gaits and perturbation conditions were collected. The mean and standard deviations were then used to center and normalize the data.

**Markov chain metric validation.** Because the methods to calculate the fall risk metrics from the Markov chain have not been used before, they were compared to the values from the brute force method. For each case and metric, the Markov chain produced a single value. In contrast, using trajectories of fifty steps for the brute force method, each metric for each case was calculated 200 times. These 200 values formed an approximately normal distribution. For a particular gait and metric, the two methods were considered equivalent if the Markov chain value was within one standard deviation of the brute force method's mean. To determine if a particular metric type could be accurately calculated from the Markov chain, the percent of gaits with equivalent values was computed. In addition, the median absolute difference between the two methods over all gaits was computed for each metric type. This indicated the magnitude of the error. For the joint variability and the Lyapunov exponents, the results for all coordinates were combined when calculating the validation statistics.

**Effect of number of steps on brute force metric values.** To further validate the Markov chains metrics and to quantify how the number of steps effected the brute force metrics, the difference between the two methods was calculated as a function of the number of steps in the brute force data. To do so, each metric was calculated using each brute force simulation. Then the absolute difference between the brute force metric value and the corresponding Markov chain metric value was found. The percentages of errors less than 0.1 and 0.5 in normalized units were found for each metric and number of steps in the brute force simulation. When calculating these percentages, data from all speeds, double support types, perturbation instants, and coordinates were combined. To estimate how many steps were needed for the brute force simulations, linear regression models were fit to the percentages

$$p = \alpha_1 \ln n + \alpha_0 \tag{10}$$

where $p$ is the percent of cases with errors less than the threshold, $n$ is the number of steps in the brute force simulation, and $\alpha_0$ and $\alpha_1$ are the regression coefficients. A separate model was found for each metric. Finally, using Eq 10, the estimated number of steps for 90% of the brute force simulations to be within the tolerance was calculated. If the model predicted less then 10 steps, the value was rounded up to 10.

## Proposed algorithms for predicting MFPT

To estimate the MFPT, fall risk prediction models were created. These fall risk models took a simulated gait, calculated the gait parameters, and predicted the MFPT. These models were

developed using the data from the Markov chain. Using the brute force simulations, the effect of the number steps on accuracy was evaluated.

**Individual metric models.**   First, a separate fall risk prediction model was created for each individual fall risk metric. The goodness of fit measures indicates how much predictive power each metric has. Because some of the gait parameters seemed to have a nonlinear relationship with fall risk, a quadratic regression was used. While a higher order of polynomial regression could be used, cursory testing showed that this exacerbated any errors in the gait parameters and caused the prediction to fluctuate wildly. Thus, the individual fall risk prediction models were

$$\hat{y} = \beta_0 + \beta_1 \rho_m + \beta_2 \rho_m^2 \tag{11}$$

where $\hat{y}$ is the natural log of the predicted MFPT, $\rho_m$ is the value of one metric, and $\beta_0 \ldots \beta_2$ are the model coefficients. The true MFPT for different gaits spanned approximately 15 orders of magnitude (Table 1) which would not work well in a regression model; using the natural log condensed the range to $2.3 - 35$. The predicted MFPT can easily be calculated with $\widehat{\text{MFPT}} = e\hat{y}$. The model coefficients were found using standard least-squares regression and the Markov chain training gait data. For the joint variability and the Lyapunov exponents, a separate model was found for each coordinate.

## Combined metric model

Because the MFPT may be dependent on multiple metrics, two regression models each using multiple metrics were created. Because Lyapunov exponents are dependent on the number of steps used in the calculation, the first model excluded Lyapunov exponents ($\hat{y}_{no}$). The second model included the long-term Lyapunov exponents ($\hat{y}_\lambda$) to evaluate if including them increases the accuracy of the fall risk model. As discussed later, short-term Lyapunov exponents are difficult to accurately calculate, so they were not included in either model.

The individual metrics likely contain redundant information. For example, the mean and standard deviation of speed, step length, and step duration are related to each other, and variability in one joint is not entirely independent from variability in the other joints. Therefore, standard principal component analysis (PCA) was used to transform the data so that each term contained as much unique information as possible [39]. The transformation was found using the Markov chain metrics; the same transformation was applied to the brute force metrics.

Similar to the individual fall risk models, a quadratic regression was used that initially included many possible linear and quadratic terms but no interaction terms:

$$\begin{aligned}\hat{y} = \quad &\beta_{0,0} + \beta_{1,1} \mathcal{X}_{\text{PCA},1} + \beta_{1,2} \mathcal{X}_{\text{PCA},1}^2 + \beta_{2,1} \mathcal{X}_{\text{PCA},2} + \beta_{2,2} \mathcal{X}_{\text{PCA},2}^2 \\ &+ \ldots + \beta_{15,1} \mathcal{X}_{\text{PCA},15} + \beta_{15,2} \mathcal{X}_{\text{PCA},15}^2,\end{aligned} \tag{12}$$

where $\hat{y}$ is the natural log of the predicted MFPT, $\mathcal{X}_{\text{PCA},1} \ldots \mathcal{X}_{\text{PCA},15}$ are the first 15 PCA scores formed by combining multiple gait metrics, and $\beta_{0,0} \ldots \beta_{15,2}$ are constant coefficients. To find the coefficients, stepwise regression and the Markov chain metrics were used. The final models were found using two steps. First, the transformed set of gait parameters $\mathcal{X}_{\text{PCA}}$ was reduced in size by 10% by removing rows corresponding to 3 testing conditions. Then a standard stepwise regression algorithm was used to determine the components of $\mathcal{X}_{\text{PCA}}$ that created the best reduced model. This was repeated 10 times to get 10 sets of components for their respective models. Next, those components present in at least 3 reduced models were collected. The stepwise algorithm was used on this reduced set of components to get the final components used

in the fall risk model. This method reduced the effect of any one testing condition on the model and reduced the chance of overfitting the model to this set of data.

**Evaluation of fall risk models.** Once the fall risk models were created, they were evaluated, first using the Markov chain metrics (i.e. the data used to create the models) and then using the brute force metrics. For the initial validation with the Markov chain metrics, the first goodness of fit measure was the adjusted $R^2$ for Eqs 11 & 12. Error was quantified using a set of goodness of fit metrics:

- The root mean square error (RMSE),

- The mean absolute error (MAE),

- The mean absolute percentage error (MAPE)

$$\text{MAPE} = \frac{100\%}{n} \sum \left| \frac{\hat{y} - y}{y} \right| \tag{13}$$

where $\hat{y}$ is the natural log of the predicted MFPT, $y$ is the natural log of true MFPT, and $n$ is the number of tested cases,

- The relative average absolute error (RAAE)

$$\text{RAAE} = \frac{\sum |\hat{y} - y|}{n \sigma \hat{y}} \tag{14}$$

where $\sigma \hat{y}$ is the standard deviation of $\hat{y}$, and

- the maximum absolute error between the natural logs of true MFPT and predicted MFPT.

In addition, the percentage of training conditions in which the predicted MFPT was within ln(MFPT)±3.3 was found. The 3.3 tolerance was 10% of the log transformed range of MFPT (Table 1). For the combined fall risk models (Eq 12), the performance was further evaluated using a linear fit relating the true and predicted MFPT:

$$\hat{y} = \alpha_1 y + \alpha_0 \tag{15}$$

where $\alpha_0$ is the intercept, and $\alpha_1$ is the slope. Ideally, the data would all fall on a line with a slope of 1 and an intercept of 0. The adjusted $R^2$ value for Eq 15 was also found. To distinguish it from the $R^2$ value for the fall risk models, it will be denoted with $R_e^2$.

The fall risk models were further validated using the brute force metrics. Essentially the same goodness of fit measures were used: MAE, MAPE, RAAE, maximum absolute error, percent of trials within tolerance, and the three measures relating to Eq 15. When computing the summary statistics, the different training speeds, double support models, perturbation timings, and trials of a particular length were combined. Separate values were calculated for each of the different simulation lengths so that the effect of the number of steps could be evaluated. In addition, the standard deviation of the error across all trails for a particular training condition and number of steps was calculated. The mean standard deviation across all training conditions was calculated.

Summary goodness of fit measures for all twelve validation conditions were also calculated.

## Results and discussion

This section first compares the gait parameter values to those in literature and discusses the validation of the Markov chain metrics. It then discusses how the number of steps in a brute force simulation affects the metrics themselves. With the individual metrics validated, the

section discusses how well they can predict the MFPT, both individually and combined together. Finally, the combined fall risk models (Eq 12) are validated using completely new gaits.

## Comparison to human gait parameter values

The mean speed, step length, and step duration matched the expected values for all cases (compare Tables 1 and 2 for an example). The step duration standard deviations were mostly on the order of 0.001 to 0.01 s. This is somewhat lower than is typically reported for human subjects (order 0.01 to 0.1 s) [25, 40–42] but somewhat higher than in another simulation study (order 0.001 s) [24]. This study's step length standard deviation (order $10^{-10}$ m) was much lower than in human experiments (order $10^{-2}$ m) [40, 42]. The very low step length standard deviation for this model was caused by the controller, because it very tightly regulated position of the actuated joints, which completely defined the step length.

For XCoM, this study found mean values on the order of 70 cm, which is substantially higher than in previous human studies [13, 14]. The Lyapunov exponent values were also substantially different than those found in human experimental studies. The experimental studies typically found short-term Lyapunov exponents on the order of 1 [18, 21, 22] while this study found values on the order of 0.01. The human studies found long-term Lyapunov exponents on the order of $10^{-2}$ [18, 21] while this study found mostly negative long-term Lyapunov exponents on the order of $10^{-4}$. One possibility for this difference is in how the data was processed. In the human studies [18, 21, 22], the entire trial was sampled at a uniform rate. In this study, each step was re-sampled to an equal number of points, which removes temporal information about step duration variability from the data.

Overall, the data in this study has less variability than is typically observed in human experimental studies. This is likely partly due to the controller, which regulates position much more tightly than humans do. It is also likely partly due to the discrete nature of the perturbations, as opposed to continuous sensory noise in a human.

## Markov chain metric validation

Overall, most of the gait parameters calculated using the Markov chain were in agreement with those calculated using the brute force method (Tables 2 and 3). The notable exception was the short-term Lyapunov exponents that almost never matched between the two methods. Ignoring the short-term Lyapunov exponents, the Markov chain values were within one standard deviation of the brute force mean in at least 50% of the cases. For approximately half of the metrics, over 80% of the cases matched. For the XCoM, FRI, walking speed, step length, and step duration, the magnitude of the error was very small, with over 50% of the differences less than 0.01 in normalized units and over 80% of the differences less than 0.1 in normalized units. The raw values were also very similar, with typical differences on the order of 0.01 or less. These differences are unlikely to be meaningfully different even if the Markov chain value is more than one standard deviation from the brute force mean. The magnitude of the error was larger for the long-term Lyapunov exponents, with only 13% of the differences less than 0.1 in normalized units. This was also seen in the raw data, with the Markov chain values tending to be at least an order of magnitude larger than the brute force values. However, over 50% of the differences were less than 1 in normalized units and almost 90% of the differences were less than 2 in normalized units. Given that the standard deviation of the brute force method was also large ($\approx$1), these larger differences are acceptable. The joint variability differences were acceptably small, with almost 50% less than 0.1 in normalized units and over 90% less than 0.5 in normalized units.

**Table 2. Un-normalized fall risk metrics for one representative case (nominal speed ≈1.0 m/s with a finite-time double support and perturbations at the beginning of double support).** The values from the Markov chain calculations and the mean ± standard deviation from the 50 step brute force simulations are given. Similar to the comparison with normalized values, most metrics matched between the two methods. The main exceptions were the Lyapunov exponents, with the brute force values consistently much smaller than the Markov chain values.

| | | Markov Chain | Brute Force |
|---|---|---|---|
| XCoM (m) | Mean | 0.672 | $0.672 \pm 0.000372$ |
| | SD | $4.39 \times 10^{-6}$ | $4.51 \times 10^{-6} \pm 1.01 \times 10^{-6}$ |
| FRI (m) | Mean | 0.0120 | $0.0118 \pm 8.39 \times 10^{-5}$ |
| | SD | $4.33 \times 10^{-8}$ | $1.41 \times 10^{-7} \pm 3.06 \times 10^{-8}$ |
| Walking Speed (m/s) | Mean | 0.979 | $0.979 \pm 0.00150$ |
| | SD | $7.15 \times 10^{-5}$ | $7.36 \times 10^{-5} \pm 1.67 \times 10^{-5}$ |
| Step Length (m) | Mean | 0.974 | $0.974 \pm 3.39 \times 10^{-6}$ |
| | SD | $1.57 \times 10^{-10}$ | $3.29 \times 10^{-10} \pm 7.86 \times 10^{-10}$ |
| Step Duration (s) | Mean | 0.994 | $0.995 \pm 0.00153$ |
| | SD | $7.38 \times 10^{-5}$ | $7.61 \times 10^{-5} \pm 1.76 \times 10^{-5}$ |
| $\lambda_S$ | 1 | 0.0264 | $0.0126 \pm 0.000886$ |
| | 2 | 0.0266 | $0.0140 \pm 0.000938$ |
| | 3 | 0.0262 | $0.0116 \pm 0.00109$ |
| | 4 | 0.0262 | $0.0115 \pm 0.00106$ |
| | 5 | 0.0276 | $0.0121 \pm 0.00105$ |
| | 6 | 0.0276 | $0.0121 \pm 0.00104$ |
| | 7 | 0.0480 | $0.0207 \pm 0.000963$ |
| | 8 | 0.0493 | $0.0265 \pm 0.00107$ |
| | 9 | 0.0303 | $0.0118 \pm 0.00103$ |
| | 10 | 0.0302 | $0.0112 \pm 0.00102$ |
| | 11 | 0.0307 | $0.0106 \pm 0.000920$ |
| | 12 | 0.0310 | $0.0108 \pm 0.000906$ |
| $\lambda_L$ | 1 | $-2.86 \times 10^{-4}$ | $-8.06 \times 10^{-6} \pm 1.25 \times 10^{-4}$ |
| | 2 | $-2.79 \times 10^{-4}$ | $-6.52 \times 10^{-6} \pm 1.29 \times 10^{-4}$ |
| | 3 | $-2.78 \times 10^{-4}$ | $7.73 \times 10^{-6} \pm 1.59 \times 10^{-4}$ |
| | 4 | $-3.11 \times 10^{-4}$ | $-4.31 \times 10^{-6} \pm 1.49 \times 10^{-4}$ |
| | 5 | $-2.92 \times 10^{-4}$ | $7.27 \times 10^{-6} \pm 1.46 \times 10^{-4}$ |
| | 6 | $-3.11 \times 10^{-4}$ | $5.04 \times 10^{-6} \pm 1.44 \times 10^{-4}$ |
| | 7 | $-8.09 \times 10^{-4}$ | $4.90 \times 10^{-4} \pm 4.92 \times 10^{-4}$ |
| | 8 | $-8.87 \times 10^{-4}$ | $4.71 \times 10^{-4} \pm 4.60 \times 10^{-4}$ |
| | 9 | $-3.06 \times 10^{-4}$ | $1.18 \times 10^{-5} \pm 1.55 \times 10^{-4}$ |
| | 10 | $-3.32 \times 10^{-4}$ | $1.06 \times 10^{-6} \pm 1.43 \times 10^{-4}$ |
| | 11 | $-3.06 \times 10^{-4}$ | $1.14 \times 10^{-5} \pm 1.36 \times 10^{-4}$ |
| | 12 | $-3.46 \times 10^{-4}$ | $-1.56 \times 10^{-6} \pm 1.37 \times 10^{-4}$ |
| Joint Angle Variability | 1 | -17.7 | $-17.7 \pm 0.238$ |
| | 2 | -15.5 | $-15.5 \pm 0.230$ |
| | 3 | -17.0 | $-17.0 \pm 0.233$ |
| | 4 | -13.4 | $-13.4 \pm 0.222$ |
| | 5 | -15.5 | $-15.5 \pm 0.232$ |
| | 6 | -14.4 | $-14.4 \pm 0.232$ |
| | 8 | -13.7 | $-14.3 \pm 1.33$ |

(*Continued*)

**Table 2.** (Continued)

| | | Markov Chain | Brute Force |
|---|---|---:|---:|
| Joint Velocity Variability | 1 | -11.2 | -11.2 ± 0.225 |
| | 2 | -9.20 | -9.20 ± 0.238 |
| | 3 | -10.4 | -10.4 ± 0.223 |
| | 4 | -7.35 | -7.34 ± 0.236 |
| | 5 | -8.81 | -8.81 ± 0.228 |
| | 6 | -7.91 | -7.91 ± 0.234 |
| | 7 | -9.67 | -9.67 ± 0.218 |
| | 8 | -13.2 | -13.2 ± 0.400 |

For the short-term Lyapunov exponents, only 1% of the values matched between the Markov chain and brute force methods. The error was also very large, with almost 50% of the differences greater than 1 in normalized units despite the typical brute force standard deviation being less than 0.1. The large differences were also present in the raw values. This is most likely due to the discretization needed to create the Markov chain because it causes a small range of actual step trajectories to map to the same transition in the Markov chain and can cause a small discontinuity between the end of one step and beginning of the next step. These discontinuities may cause errors when measuring differences in the joint trajectories over the duration of two steps. While it may be possible to refine the Markov chain by reducing the discretization threshold distance used to create it, this is computationally expensive and would eventually become intractable. The long-term Lyapunov exponents calculated with the Markov chain seem to be much more accurate than the short-term Lyapunov exponents. This is expected if the issue with short-term Lyapunov exponents was due to the discretization of the Markov chain. Because the long-term Lyapunov exponents evaluate how neighboring trajectories behave over the duration of 4 to 10 strides, any discontinuities between the individual steps become much less significant than the perturbations in the gait.

**Table 3. Comparison of fall risk metrics between the Markov chain and brute force methods.** The % Match column gives the percentage of Markov chain values that were within one standard deviation of the corresponding brute force values. The median error column give the median of the absolute difference between methods. The BF SD column gives the median of the standard deviations from the brute force method. Most metrics matched between the two methods most of the time, with the notable exception of the short-term Lyapunov exponents.

| | | % Match | Median Err | BF |
|---|---|---:|---:|---:|
| XCoM | Mean | 93 | 0.00918 | 0.0573 |
| | SD | 90 | 0.160 | 0.239 |
| FRI | Mean | 60 | 0.00713 | 0.0113 |
| | SD | 50 | 0.00320 | 0.00759 |
| Walking Speed | Mean | 90 | 0.0198 | 0.116 |
| | SD | 77 | 0.0613 | 0.165 |
| Step Length | Mean | 67 | $3.37 \times 10^{-5}$ | $9.43 \times 10^{-5}$ |
| | SD | 80 | 0.00218 | 0.0124 |
| Step Duration | Mean | 90 | 0.0162 | 0.126 |
| | SD | 83 | 0.0274 | 0.0710 |
| $\lambda_S$ | | 1 | 0.955 | 0.0981 |
| $\lambda_L$ | | 67 | 0.671 | 1.01 |
| Joint Variability | Angle | 71 | 0.154 | 0.233 |
| | Velocity | 68 | 0.113 | 0.214 |

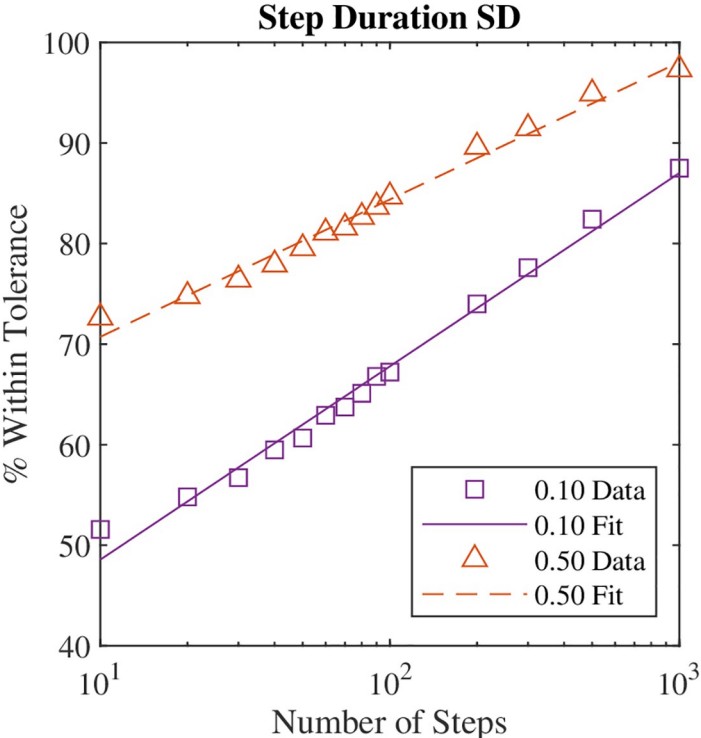

**Fig 2. Brute force simulation accuracy vs. number of steps in the simulation.** The percentage of simulations within 0.1 or 0.5 normalized units of the Markov chain metric values is shown. As the number of steps in the brute force simulation increases, more simulations match the Markov chain step duration standard deviation. The increase in accuracy as the number of steps in the brute force simulation increases is true for all gait parameters.

Therefore, the method developed in this paper to calculate the fall risk metrics seems to produce values in agreement with the standard calculation method for all tested variables except for the short-term Lyapunov exponents.

### Effect of number of steps on brute force metric values

As expected, the difference between the metrics calculated using the Markov chain and the brute force simulations decreased as the number of steps in the brute force simulation increased (Fig 2). This was true for all metrics. The linear regression line used to estimate the number of steps required (Eq 10) fit the data very well in most cases, with $R^2$ values consistently above 0.97. This indicates that the brute force values are converging to the Markov chain values, and thus, that the two methods are capturing fundamentally the same gait parameters. The major exception was the long-term Lyapunov exponent; the percentage of trials within the threshold appeared to plateau at approximately 100 steps, so the data did not fall along a line. Interestingly, this behavior was not observed for the short-term Lyapunov exponents.

In order for 90% of the brute force trials to have a metric value within 0.1 normalized units of the Markov chain value, the brute force simulation generally required several thousand steps (Table 4). The major exceptions were the Lyapunov exponents and step length mean and standard deviation. The step length metrics could be accurately estimated using 10 steps. This is likely a simulation artifact because the control method very tightly regulated step length, so

**Table 4. Estimated number of steps required for 90% of the brute force metric values to be within 0.1 or 0.5 normalized units of the Markov chain metric values.**

| Threshold | | 0.1 | 0.5 |
|---|---|---:|---:|
| XCoM | Mean | 322 | 10 |
| | SD | 28,600 | 413 |
| FRI | Mean | 10 | 10 |
| | SD | 26,000 | 10 |
| Walking Speed | Mean | 2,320 | 70 |
| | SD | 9,720 | 334 |
| Step Length | Mean | 10 | 10 |
| | SD | 12 | 10 |
| Step Duration | Mean | 1,860 | 74 |
| | SD | 1,430 | 258 |
| $\lambda_S$ | | $2.68 \times 10^{13}$ | 31,400 |
| $\lambda_L$ | | $3.37 \times 10^{17}$ | 219,000 |
| Joint Variability | Angle | 3,650 | 330 |
| | Velocity | 2,070 | 113 |

all values were very tightly clustered around the mean value for that particular gait. Previous work using experimental data to accurately estimate stride length and/or duration found that several hundred strides were needed [43, 44]. Thus, this work's estimates are different by an order of magnitude. This difference is partly due to the definition of accurate enough, which is not standardized across studies. This work used normalized values to define accuracy, and if the scaling factor was small, the difference in dimensional values may be insignificant even if the difference in normalized values is relatively large. For example, for mean step duration, a normalized value of 0.5 corresponds to a dimensional value of 0.08 s, which for many practical cases, may be sufficiently accurate.

According to the linear regression model, both Lyapunov exponents would require an excessive number of steps in the brute force simulation to accurately estimate the value. It is unlikely that the estimated number of steps are accurate given the very large extrapolation required. Further, the Markov chain Lyapunov exponent values may not be accurate, particularly for the short-term Lyapunov exponents. However, it is highly likely that calculating the Lyapunov exponents requires far more steps than any of the other measures. This largely agrees with previous literature on human subject experiments. [21] found that increasing the number of steps changed the estimated short- and long-term Lyapunov values up to their maximum trial length of 300 strides. [45] also found that the short-term Lyapunov exponents did not fully plateau within a 5 min trial (about 300 strides), but that the long-term Lyapunov exponent values could be accurately estimated using a 3 min (about 200 strides) trial. In contrast, [44] estimated that about 100 strides were needed to calculate both the short- and long-term Lyapunov exponents with a 10% error. Thus, it seems possible that well over one thousand steps are required to accurately calculate Lyapunov exponents.

## Individual metric fall risk prediction models

None of the fall prediction models using a single metric (Eq 11) performed well, even in the best-case scenario of evaluating performance using the data used to create the model. The $R^2$ values were consistently very low, with 26 of the 49 models having an $R^2$ value less than 0.1, and 45 of the 49 models having an $R^2$ value less than 0.3. None of the models had an $R^2$ value

above 0.5. The joint variability models generally had the highest $R^2$ values—they ranged from 0.06 to 0.46. Given the overall poor fit, the errors were very large. The mean errors ranged from 5.58 to 8.73 for log-transformed MFPT, with most (42 out of 49 or 86%) models having mean errors greater than 7. Not surprisingly, few of the individual predictions were within the 10% tolerance; most (32 out of 49 or 65%) models had between 20 and 40% of its predictions within the tolerance band. No model had more than 50% of its predictions within the tolerance band.

Given that the Markov chain and brute force metrics were similar, the individual models had similar poor performance when tested using the brute force data. 98% of the cases had mean errors greater than 7. Essentially all (99%) cases had between 5 and 35% of the individual predictions within the tolerance band. For some of the Lyapunov exponents, the error decreased as the number of steps used to calculate the metric increased. For the other metrics, the number of steps used to estimate the metric did not substantially change the error. However, for very low numbers of steps (less than about 50), the percentage of predictions within the tolerance band increased by 5 to 10% as more steps were used to calculate the metric. Above 50 steps, the percentage within the tolerance band was insensitive to the number of steps used. Thus, none of the metrics by themselves could predict MFPT.

## Combined metric fall risk prediction models

The combined fall risk models performed significantly better than the individual fall risk models (Fig 3, Table 5). For the Markov chain data, the mean errors, given as the absolute

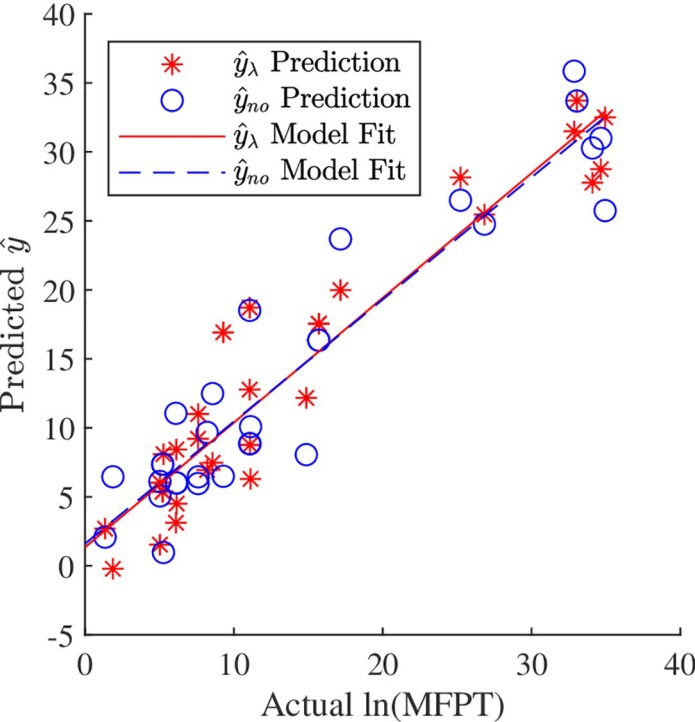

**Fig 3. Fall risk model prediction ($\hat{y}$) compared to the true MFPT for the Markov chain data used to create the model.** The two lines on the plot are a linear fit of the data (Eq 15). In general, as the actual MFPT increased, the prediction from both models increased.

**Table 5. Summary statistics for the fall risk models using data from the Markov chains.** $R^2$ gives the adjusted $R^2$ value for the fitted model (Eq 12). The root mean squared error (RMSE), mean absolute error (MAE), mean absolute percentage error (MAPE), relative average absolute error (RAAE), and maximum absolute error (Max Abs Err) are over all training conditions. The % in Tolerance indicates the percentage of these trials in which the predicted value was within ln(MFPT)±3.3. The slope, intercept, and $R_e^2$ are from Eq 15. Values of 1, 0, and 1, respectively, would indicate a perfect fit.

| Fall Risk Model | $\hat{\mathbf{y}}_{no}$ | $\hat{\mathbf{y}}_\lambda$ |
|---|---|---|
| $R^2$ | 0.849 | 0.867 |
| RMSE | 4.20 | 3.94 |
| MAE | 2.75 | 2.66 |
| MAPE | 32.7 | 30.7 |
| RAAE | 0.271 | 0.259 |
| Max Abs Err | 9.20 | 7.65 |
| % in Tolerance | 66.7 | 76.7 |
| Slope | 0.885 | 0.904 |
| Intercept | 1.60 | 1.34 |
| $R_e^2$ | 0.881 | 0.900 |

difference between $\hat{y}$ and the natural log of the MFPT, were between 2.5−3 and the maximum errors were from 7−10, with the fall risk model that excludes long-term Lyapunov exponents having slightly higher errors. For this data, both models predicted MFPT within tolerance for ≈70% of the trials. In general, as the true MFPT increased, the prediction also increased. For the linear fit between the predicted and true ln(MFPT) (Eq 15), both fall risk models had slopes close to 1. As with the other goodness of fit measures, the fall risk model that includes long-term Lyapunov exponents was slightly closer with a slope of 0.904 and an intercept of 1.34. The data was also slightly more tightly clustered along the line for the model that includes long-term Lyapunov exponents than for the model that excludes the long-term Lyapunov exponents ($R_e^2$ of 0.900 vs. 0.881).

As expected, the stepwise regression removed many possible terms from the regression model (Eq 12, Table 6). For the model that excludes the long-term Lyapunov exponents, the final model had 8 terms out of a possible 31. For the model that includes the long-term Lyapunov exponents, the final model had 9 terms out of a possible 31. The drastic reduction in the number of terms confirms that the tested metrics contain a significant amount of redundant information and/or information that is not related to fall risk. The terms in the model did not strictly correspond to the first few principal components, although both models included several terms relating to the first few principal components. In general, the principle components did not closely align with any individual gait parameter or easily interpreted group of parameters. Thus, both models appeared to use information from all of the gait parameters used to create the PCA components.

## Effect of the number of steps on the combined fall risk models

As expected, the fall risk models also performed well using the brute force data for the longer simulations (Table 7). For the model that excludes the long-term Lyapunov exponents, the MAE was minimized at 4.71 when 500 steps were used, and the maximum error was minimized at 17.1 when 1000 steps were used. For the model that includes the long-term Lyapunov exponents, the MAE was minimized at 5.45 when 1000 steps were used, and the maximum error was minimized at 15.9 when 1000 steps were used. These errors were somewhat higher

**Table 6. Fall risk model parameters.** Parameters not listed on these tables have a value of 0.

| Parameter | Value | Component |
|---|---|---|
| (**a**) Fall risk model excluding long-term Lyapunov exponents. | | |
| $\beta_{0,0}$ | 6.28 | — |
| $\beta_{4,2}$ | -2.59 | $\mathcal{X}^2_{no,PCA,4}$ |
| $\beta_{5,1}$ | -9.51 | $\mathcal{X}_{no,PCA,5}$ |
| $\beta_{5,2}$ | 13.6 | $\mathcal{X}^2_{no,PCA,5}$ |
| $\beta_{9,1}$ | -14.5 | $\mathcal{X}_{no,PCA,9}$ |
| $\beta_{10,1}$ | 7.77 | $\mathcal{X}_{no,PCA,10}$ |
| $\beta_{11,2}$ | 45.2 | $\mathcal{X}^2_{no,PCA,11}$ |
| $\beta_{13,1}$ | -21.7 | $\mathcal{X}_{no,PCA,13}$ |
| (**b**) Fall risk model including long-term Lyapunov exponents. | | |
| $\beta_{0,0}$ | 22.4 | — |
| $\beta_{1,1}$ | -1.46 | $\mathcal{X}_{\lambda,PCA,1}$ |
| $\beta_{1,2}$ | -0.413 | $\mathcal{X}^2_{\lambda,PCA,1}$ |
| $\beta_{2,1}$ | 1.62 | $\mathcal{X}_{\lambda,PCA,2}$ |
| $\beta_{3,1}$ | -2.00 | $\mathcal{X}_{\lambda,PCA,3}$ |
| $\beta_{9,1}$ | 5.27 | $\mathcal{X}_{\lambda,PCA,9}$ |
| $\beta_{9,2}$ | -9.18 | $\mathcal{X}^2_{\lambda,PCA,9}$ |
| $\beta_{12,1}$ | -6.83 | $\mathcal{X}_{\lambda,PCA,12}$ |
| $\beta_{13,2}$ | 36.6 | $\mathcal{X}^2_{\lambda,PCA,13}$ |

than for the Markov chain metrics, but this was expected since the models were created using the Markov chain metrics. The percent of trials within the tolerance of 3.3 was also somewhat lower than for the Markov chain data. For the model that excludes the long-term Lyapunov exponents, the percentage was maximized at 41.6% when 300 steps were used, and for the model that includes the long-term Lyapunov exponents, the percentage was maximized at 32.1% when 1000 steps were used. Even for the longest trials of 1000 steps, different trials gave different predictions. The mean of the standard deviations reached a minimum of approximately 5 at 1000 steps for both models. The linear fit between the predicted and true ln (MFPT) (Eq 15) for the model that excludes the long-term Lyapunov exponents was of similar quality to the one from the Markov chain results. When 1000 steps were used, the slope was 0.8 (ideal value was 1), the intercept was 6 (ideal values was 0), and $R^2_e$ was 0.7 (ideal value was 1). The goodness of fit was somewhat lower for the model that includes the long-term Lyapunov exponents, with a maximum slope of 0.7, minimum intercept of 7, and $R^2_e$ of 0.6, all when 1000 steps were used. Thus, while neither model reach the accuracy achieved when using the data that created the model, the results were acceptable when large numbers of steps were used.

For both models, the prediction generally became more accurate as the number of steps used increased. For the most part, RMSE, MAE, MAPE, maximum absolute error, and mean of the standard deviations all decreased as the number of steps increased for both models. Similarly, the percentage of trials within tolerance increased as the number of steps increased. The goodness of fit between the predicted and true ln(MFPT) (Eq 15) also increased as the number of steps increased. The slopes for both models increased toward 1, the intercept decreased toward 0, and $R^2_e$ increased as the number of steps increased. The slope in Eq 15 was positive

**Table 7. Summary statistics for both combined fall risk models as the number of steps used changes.** $\hat{y}_{no}$ indicates the model that excludes the long-term Lyapunov exponents, and $\hat{y}_{\lambda}$ indicates the model that includes this data. Each column indicates the number of steps used in the fall risk model. For each number of steps $n$, a simulation was created (usually taken from a longer simulation), fall risk metrics were calculated from this simulation, the PCA transformation was applied, and then ln(MFPT) was estimated using Eq 12. The values are over all of the trials of a given number of steps $n$. The root mean squared error (RMSE), mean absolute error (MAE), mean absolute percentage error (MAPE), relative average absolute error (RAAE), and maximum absolute error (Max Abs Err) are given. The % in Tolerance indicates the percentage of these trials in which the predicted value was within the true ln(MFPT)±3.3. The mean SD are the mean of the standard deviation data for each training condition. The slope, intercept, and $R_e^2$ are each from a linear fit of the true and estimated ln(MFPT) for all trials (Eq 15). Values of 1, 0, and 1, respectively, would indicate a perfect fit. Cells with no value indicate that there was not enough data to calculate these statistics for the given number of steps.

| | Number of Steps Used | | | | | | | | | | | | | |
|---|---|---|---|---|---|---|---|---|---|---|---|---|---|---|
| $\hat{y}_{no}$ | **10** | **20** | **30** | **40** | **50** | **60** | **70** | **80** | **90** | **100** | **200** | **300** | **500** | **1000** |
| **RMSE** | 7,630 | 2,810 | 1,820 | 1,500 | 1,220 | 1,070 | 888 | 697 | 606 | 550 | 252 | 182 | 130 | 88.5 |
| **MAE** | 13.2 | 11 | 9.91 | 9.03 | 8.52 | 8.06 | 7.55 | 7.04 | 6.67 | 6.58 | 5.21 | 5.04 | 4.71 | 4.73 |
| **MAPE** | 324 | 247 | 208 | 178 | 159 | 147 | 127 | 112 | 102 | 95.8 | 54.2 | 46.6 | 40.5 | 38.3 |
| **RAAE** | 0.322 | 0.53 | 0.581 | 0.535 | 0.537 | 0.503 | 0.506 | 0.522 | 0.518 | 0.522 | 0.477 | 0.47 | 0.458 | 0.486 |
| **Max Abs Err** | 4,420 | 958 | 336 | 271 | 233 | 352 | 262 | 184 | 160 | 139 | 35.5 | 29 | 18.6 | 17.1 |
| **% in Tolerance** | 23.1 | 26.5 | 29.3 | 32.5 | 33.5 | 35.4 | 35.9 | 37.1 | 37.8 | 38 | 40.5 | 41.6 | 41.5 | 39.3 |
| **Mean SD** | 41.3 | 20.7 | 16.4 | 15.9 | 14.6 | 14.6 | 13 | 10.9 | 10 | 9.55 | 6.41 | 6.06 | 5.47 | 5.41 |
| **Slope** | 0.355 | 0.518 | 0.598 | 0.646 | 0.678 | 0.7 | 0.744 | 0.782 | 0.794 | 0.806 | 0.853 | 0.856 | 0.847 | 0.799 |
| **Intercept** | 20 | 15.6 | 13.4 | 11.8 | 10.8 | 10 | 8.93 | 7.97 | 7.36 | 7.1 | 5.26 | 5.05 | 4.98 | 5.86 |
| $R_e^2$ | 0.0082 | 0.0681 | 0.135 | 0.161 | 0.201 | 0.211 | 0.274 | 0.372 | 0.42 | 0.452 | 0.674 | 0.7 | 0.741 | 0.736 |
| $\hat{y}_{\lambda}$ | **10** | **20** | **30** | **40** | **50** | **60** | **70** | **80** | **90** | **100** | **200** | **300** | **500** | **1000** |
| **RMSE** | – | — | 5,010 | 2,670 | 1,900 | 1,430 | 1,220 | 956 | 845 | 765 | 370 | 249 | 175 | 96.2 |
| **MAE** | — | — | 28.3 | 19 | 15.6 | 13.7 | 12.6 | 11.3 | 10.6 | 10.4 | 7.79 | 7.08 | 6.3 | 5.45 |
| **MAPE** | — | — | 457 | 332 | 275 | 245 | 219 | 194 | 179 | 169 | 95.8 | 76.8 | 63.4 | 50.6 |
| **RAAE** | — | — | 0.664 | 0.77 | 0.783 | 0.812 | 0.808 | 0.859 | 0.842 | 0.851 | 0.759 | 0.71 | 0.674 | 0.619 |
| **Max Abs Err** | — | — | 703 | 495 | 348 | 318 | 233 | 148 | 94.5 | 85.1 | 60.6 | 39 | 25.2 | 15.9 |
| **% in Tolerance** | — | — | 10.8 | 14.4 | 15.6 | 18.9 | 20 | 22.5 | 23.5 | 22.8 | 28.1 | 28 | 30.1 | 32.1 |
| **Mean SD** | — | — | 42.9 | 25.7 | 20.8 | 17.4 | 16.2 | 13.5 | 12.8 | 12.1 | 9.16 | 8.33 | 7.58 | 6.3 |
| **Slope** | — | — | 0.394 | 0.265 | 0.336 | 0.416 | 0.415 | 0.462 | 0.481 | 0.508 | 0.598 | 0.636 | 0.637 | 0.674 |
| **Intercept** | — | — | 34.3 | 27.4 | 22.9 | 19.8 | 18.8 | 17.1 | 15.9 | 15.3 | 11.1 | 9.47 | 8.75 | 6.97 |
| $R_e^2$ | — | — | 0.00936 | 0.0125 | 0.0312 | 0.0668 | 0.0777 | 0.136 | 0.16 | 0.191 | 0.373 | 0.447 | 0.507 | 0.637 |

for both models regardless of the number of steps used, indicating that the predicted number of steps increased as the actual MFPT increased. However, when low numbers of steps were used ($\leq 100$), $R_e^2$ was very low and the standard deviation between trials was very large. Thus, for low numbers of steps, a single prediction from the fall risk model is not sufficient to estimate the MFPT or even compare the MFPT of two gaits. In other words, using less than 100 steps in the fall risk model will not yield useful predictions. Using 1000 or more steps seems to result in the best prediction. However, using so many steps may be infeasible for some gaits. As seen in Table 1, several gaits were unable to walk continuously for 1000 or even 500 steps. Thus, in order to make these fall risk models as broadly applicable as possible, using 200–300 steps seems to be the best tradeoff between accuracy and usefulness. With the exception of a few egregious outliers, the fall risk models fit moderately well with the brute force data using 300 steps (Fig 4a).

## Effect of Lyapunov exponents

In general, it appears that including Lyapunov exponents did not substantially increase model accuracy, and in some cases, resulted in much greater errors in the predicted values (Tables 5 & 7). For the brute force data, the model that excludes the Lyapunov exponents generally had

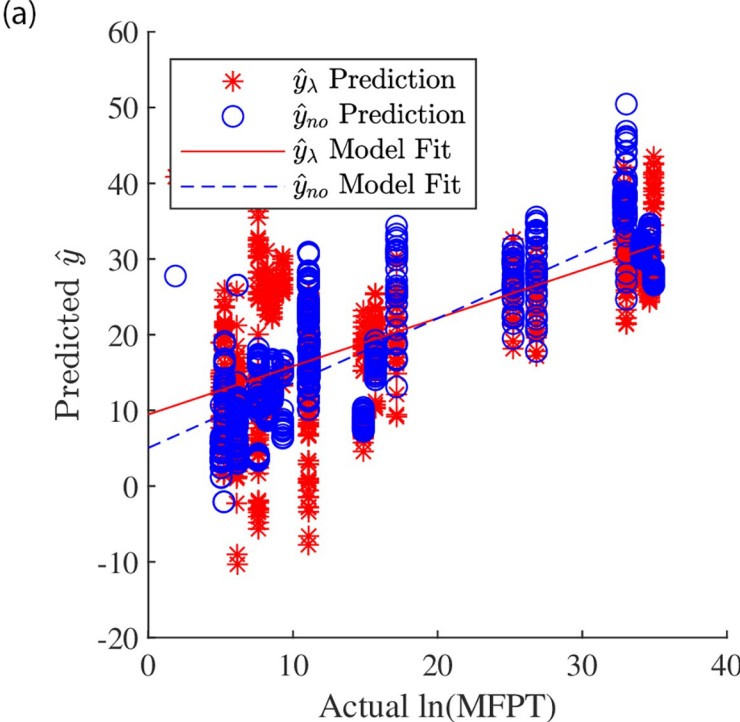

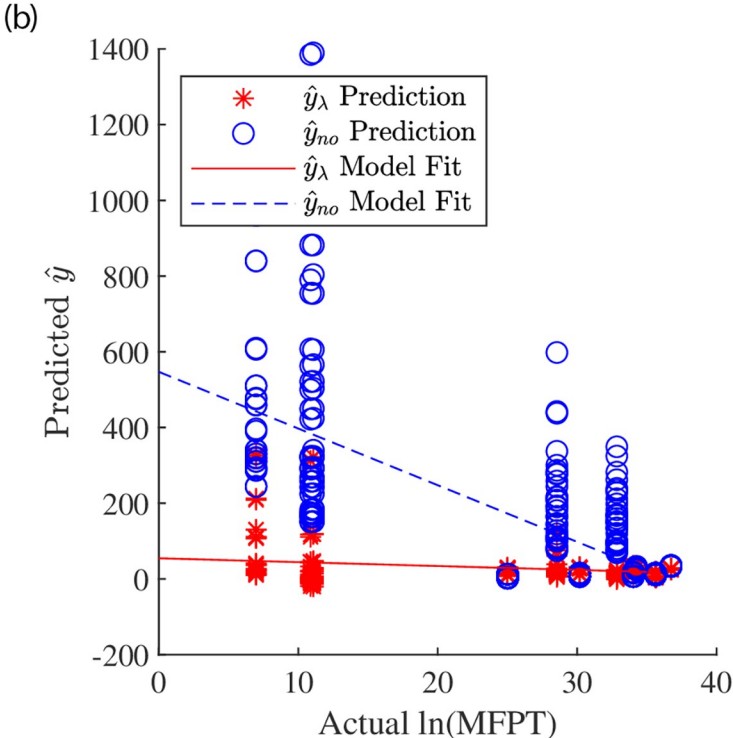

**Fig 4. Fall risk model prediction ($\hat{y}$) compared to the actual MFPT for simulations with 300 steps.** (a) shows results for the training gaits that were used to create the model, and (b) shows results for the validation gaits that the model had never seen. Red stars indicate the prediction using the fall risk model that includes the long-term Lyapunov exponents ($\hat{y}_\lambda$) and blue circles indicate the prediction using the fall risk model that excludes the long-term Lyapunov exponents ($\hat{y}_{no}$). In general, as the actual MFPT increased, the prediction from both models increased for both sets of data. However, the validation gaits had substantially more outliers.

better goodness of fit metrics. Specifically, it could explain more of the variance in the MFPT ($R_e^2$ was generally higher and the mean standard deviations were generally lower), the slope and intercept in Eq 15 were both closer to the ideal values, both the mean and maximum error were generally lower, and the percentage of gaits within tolerance were generally higher. In contrast, the model that includes the long-term Lyapunov exponents performed slightly better for the Markov chain data. The difference between the two data types may partially be explained by the lack of agreement between the Markov chain and brute force long-term Lyapunov exponents. While some retrospective human-subject studies have found correlations between Lyapunov exponents and fallers [7, 22], the results of this study only partially support using Lyapunov exponents to quantify fall risk.

## Model validation and discussion

As a final analysis of the two combined fall risk models, they were validated with four new gaits (the validation gaits in Table 1) and brute force simulations of 300 steps. Both models were substantially less accurate using this new validation data (Table 8, Fig 4b). This was partially expected, because the previously evaluated gaits were used to create the fall risk models. When comparing the predicted to true MFPT (Eq 15), $R_e^2$ was 0.494 for the model that excludes the long-term Lyapunov exponents and 0.060 for the model that includes the long-term Lyapunov exponents. For both models, $R_e^2$ was lower than for the training data and could therefore explain less variance in the data. Further, the slope and intercept were much further away from the desired values of 1 and 0, respectively. Similarly, the mean and maximum error were higher than the results using the previous data. Similar to the other metrics, the standard deviations were also larger than for the previous data, indicating that the predictions from different simulations were less consistent.

The results for these new validation gaits had much higher variability and less accuracy than the results for the previous training gaits. One possibility is that the fall risk metrics for these gaits were more variable than the previous gaits. This would then cause the fall risk predictions that use these metrics to become more variable. If that were the case, other validation gaits could result in less variable predictions. However, these models would be limited to less

**Table 8. Summary statistics for both fall risk models using data from the validation gaits.** $\hat{y}_{no}$ indicates the model that excludes the long-term Lyapunov exponents, and $\hat{y}_\lambda$ indicates that the model includes this data. The values are over all of the trials. The root mean squared error (RMSE), mean absolute error (MAE), mean absolute percentage error (MAPE), relative average absolute error (RAAE), and maximum absolute error (Max Abs Err) are given. The % in Tolerance indicates the percentage of these trials in which the predicted value was within the true ln(MFPT)±3.3. The mean SD are the mean of the standard deviation data for each validation condition. The slope, intercept, and $R_e^2$ are each from a linear fit of the ln(MFPT) for all validation trials (Eq 15). Values of 1, 0, and 1, respectively, would indicate a perfect fit.

| Fall Risk Model | $\hat{y}_{no}$ | $\hat{y}_\lambda$ |
|---|---|---|
| RMSE | 4,630 | 821 |
| MAE | 141 | 21.1 |
| MAPE | 1,270 | 168 |
| RAAE | 0.646 | 0.501 |
| Max Abs Err | 1,380 | 319 |
| % in Tolerance | 9.91 | 18.9 |
| Mean SD | 226 | 45.8 |
| Slope | -15 | -1.03 |
| Intercept | 547 | 54.5 |
| $R_e^2$ | 0.494 | 0.0604 |

variable gaits. Another possibility is that these two models were overfit to the original data. This would then cause new gaits, especially those more dissimilar to the original gaits, to yield less accurate predictions. However, while overfitting may explain the increased error, it does not fully explain the increase in variability.

## Conclusion

A wide range of gait metrics potentially correlated with fall risk were tested. None of these fall risk metrics could accurately predict MFPT, or the average number of steps taken before falling, by themselves. However, when many metrics were combined into a quadratic fall risk model, the predictive capability increased substantially. As expected, as the number of steps used to calculate the fall risk metrics increased, the accuracy and precision increased. This led to a corresponding increase in the accuracy and precision of the fall risk model. 300 step simulations seemed to provide the best tradeoff between accuracy and using as few steps as possible.

To calculate the ground truth MFPT and create the fall risk models, the gaits were modeled as a Markov chain. New methods were developed and validated to calculate the individual fall risk metrics from the Markov chain. The Markov chain could accurately calculate all of the fall risk metrics except the short-term Lyapunov exponents.

## Supporting information

**S1 File. Data.** File containing the data used in this work.
(ZIP)

## Author Contributions

**Conceptualization:** Daniel Williams, Anne E. Martin.

**Formal analysis:** Daniel Williams.

**Funding acquisition:** Anne E. Martin.

**Investigation:** Daniel Williams, Anne E. Martin.

**Methodology:** Daniel Williams, Anne E. Martin.

**Supervision:** Anne E. Martin.

**Writing – original draft:** Daniel Williams.

**Writing – review & editing:** Anne E. Martin.

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
