## [Decision Letter · Decision Letter 0]

26 Sep 2022

PONE-D-22-21276Predicting fall risk using multiple mechanics-based metrics for a planar biped modelPLOS ONE

Dear Dr. Martin,

Thank you for submitting your manuscript to PLOS ONE. After careful consideration, we feel that it has merit but does not fully meet PLOS ONE’s publication criteria as it currently stands. Therefore, we invite you to submit a revised version of the manuscript that addresses the points raised during the review process.

This is an interesting paper – thank you for your submission.

Please carefully address the suggestions of reviewer 1.

Reviewer 2 made some suggestions I would not agree with. E.g. in my opinion it is not necessary to split the introduction into an extra literature review. Also, the suggested additional citations are optional, don’t feel pressured to cite something you feel is not necessary.

We look forward to receiving your revised manuscript.

Kind regards,

Peter Andreas Federolf

Academic Editor

PLOS ONE

Journal Requirements:

2. Please note that PLOS ONE has specific guidelines on code sharing for submissions in which author-generated code underpins the findings in the manuscript. In these cases, all author-generated code must be made available without restrictions upon publication of the work. Please review our guidelines at https://journals.plos.org/plosone/s/materials-and-software-sharing#loc-sharing-code and ensure that your code is shared in a way that follows best practice and facilitates reproducibility and reuse. New software must comply with the Open Source Definition.

Reviewers' comments:

Reviewer's Responses to Questions

**Comments to the Author**

1. Is the manuscript technically sound, and do the data support the conclusions?

Reviewer #1: Partly

Reviewer #2: Yes

2. Has the statistical analysis been performed appropriately and rigorously? 

Reviewer #1: Yes

Reviewer #2: Yes

3. Have the authors made all data underlying the findings in their manuscript fully available?

Reviewer #1: Yes

Reviewer #2: Yes

4. Is the manuscript presented in an intelligible fashion and written in standard English?

Reviewer #1: Yes

Reviewer #2: Yes

5. Review Comments to the Author

Reviewer #1: Review: PONE-D-22-21276

Title: Predicting fall risk using multiple mechanics-based metrics for a planar biped model

This manuscript describes a study in which fall risk is being predicted by metrics calculated from simulations of a model, and from Markov chain simulations. The fall risk is expressed as the MFPT from the Markov chain. Results showed that many strides were needed to get reliable estimates of most measures took many strides, and that single parameters could not predict falls.

Overall, this is interesting work. However, I have the feeling that the authors want too much in this manuscript, which makes everything rather hard to follow. It seems like there are almost 2-3 stories in here; 1. Using Markov chains to calculate gait metrics 2. Effects of length on gait metrics 3. Fall prediction models. By combining these three, I feel that the authors do not really do justice to all of them. In addition, the authors may wish to think about the audience they are writing for. I myself am a biomechanist with (quite) some technical training, but still had quite the hard time following the parts on the Markov chains; I am no expert on this, and details were hard to follow. Hence, I will largely refrain from comments on this. Below are more detailed comments.

Major

1) The models from multiple metrics are nice, but in a way, not very informative. That is, the reader has no clue which metrics are included in the models, and, as such, the models are very hard to judge. It would be good to have at least a table with which metrics load on which PC, and then, one would ideally also like to know how all the PC’s contribute to the model.

2) It seems that for the calculation of all parameters, all stride-data is resampled to contain 100 samples. Note that for the Lyapunov exponents, this is not usually done, as usually, the temporal variations in the time series are retained (e.g. resampling 200 strides to 20000 samples, while retaining variations in amount of samples per stride). This could have serious effects on the calculated Lyapunove exponents (and their correlation with mfpt.

3) Nowhere in the manuscript can the reader find the actual values of any of the outcome parameters, making it very hard to check whether any of the findings make some sense. For instance, for the Lyapunov exponents (see minor comment 3).

4) As said, I am no expert on Markov chains, but I do find it surprising that so many strides are required to get to the “true” value of the Markov chain simulation. Could this have to do with the fact that in fact, these simulations are different in nature, so that actually, differences are to be expected? Could the authors for instance show that the variance of estimates (or the mean of the estimated) of say mean walking speed, which requires 2000+ strides to be 0.1 unit within the Markov chain simulations, indeed changes over such a long period? I would assume that a simulation of 1000 strides would not lead to a significantly different estimated walking speed from one of say 1500 strides?

Minor

1) Line 88 “without slip”, line 89 “fail by slipping” how is this possible, giving line 88?

2) It is unclear how the brute force simulations were created. The text talks of “initial set of 10 random perturbation vectors”, but the siimulations were for 100 steps. Does this mean not every step is perturbed?

3) (this could be major?); line 157-158: “Long term Lyapunov exponents are usually 2 orders of magnitude larger than short term Lyapynov exponents”; no, they are not. Actually, they are usually 2 orders of magnitude smaller. A table with actual values of the metrics (see major comment 3) would help here. Edit; I checked the data (would be nice to also have raw values in the bruteforcemetrics, and not only normalized values), and it seems that indeed the short term is an order of magnitude largerd than the short term?

4) Formula 12 misses a quadratic term for PCA2?

5) Line 302 states that Xpca are the transformed gait metrics, but it would seem that these are instead the PC scores formed from (Several) gait metrics?

6) It seems that the m/2 cannot be right in formula 12? As for m=3 this would lead to beta3 and XPCA,1.5. I think this should be floor(m/2), or some other operation.

Reviewer #2: Dear Authors,

This paper has proposed a predicting model to predict fall risk using multiple mechanics-based metrics for a planar biped model

However, authors need to clarify the following doubts.

1) In the abstract section, I think it will be wise to talk about directly about Falls in case of Robots only, no need to bring the human factor. Readers might get confused.

2) In the abstract section, you mentioned “Many mechanics-based fall risk metrics have been proposed and validated to varying degrees, including the extrapolated centre of mass, the foot rotation index, Lyapunov exponents, joint and spatiotemporal variability, and mean spatiotemporal parameters”.

Is it only about metrics? What is your proposed model that you must emphasis on abstract rather than metrics.??

3) Introduction and literature survey should be as made separate paragraphs. First , in the introduction section introduce your problem statement with several recent literature and may be few existing available solution you can mentions(as citations). The at the end write 4 to 5 objectives that you have achieved in this work.

3) After introduction, you can have 2. Literature review: where you talk about the available solutions as citations(recent)

That means

Introduction

Literature Review.

4) In the materials and methods talk about the materials first then method(proposed one i.e Biped Model). First you write about the materials i.e about the data sets.

5) In the method section write Algorithms(proposed).

6) Figure 2 : Actual vs predicted on test data or train that you mention, there is no point having actual vs predicted for train data.

7) Which ever following errors you can find, please add them to your work

variance accounted for (VAF),

relative average absolute error (RAAE),

root means absolute error (RMAE),

coefficient of determination (R2),

standard deviation ratio (RSR),

mean absolute percentage error (MAPE), Nash–Sutcliffe coefficient (NS), root means squared error (RMSE),

weighted mean absolute percent error (WMAPE)

and mean absolute percentage Error (MAPE

8) Clarify the figure 3 also(is it for test set or train set?)

9) Significance of Piped model not written on the manuscript, please talk about the significance of choosing the models(proposed) for the work, in what sense they better than the existing ones. Also do a comparative study with the existing model in other paper.

9) You must cite the following papers in the introduction and literature sections.

A) Dinegdae, Y. H., Onifade, I., Jelagin, D., & Birgisson, B. (2015). Mechanics-based top-down fatigue cracking initiation prediction framework for asphalt pavements. Road Materials and Pavement Design, 16(4), 907-927.

B) Roy, S. S., Roy, R., & Balas, V. E. (2018). Estimating heating load in buildings using multivariate adaptive regression splines, extreme learning machine, a hybrid model of MARS and ELM. Renewable and Sustainable Energy Reviews, 82, 4256-4268.

C) Robertson, S. W., & Ritchie, R. O. (2008). A fracture‐mechanics‐based approach to fracture control in biomedical devices manufactured from superelastic Nitinol tube. Journal of Biomedical Materials Research Part B: Applied Biomaterials, 84(1), 26-33.

D) Samui, P., Roy, S. S., & Balas, V. E. (Eds.). (2017). Handbook of neural computation. Academic Press.

E) Mitros, Z., Sadati, S. H., Henry, R., Da Cruz, L., & Bergeles, C. (2022). From theoretical work to clinical translation: Progress in concentric tube robots. Annual Review of Control, Robotics, and Autonomous Systems, 5, 335-359.

F) Roy, S. S., Samui, P., Deo, R., & Ntalampiras, S. (Eds.). (2018). Big data in engineering applications (Vol. 44). Berlin/Heidelberg, Germany: Springer.

6. PLOS authors have the option to publish the peer review history of their article (what does this mean?). If published, this will include your full peer review and any attached files.

Reviewer #1: **Yes: **Sjoerd Bruijn

Reviewer #2: No

---

## [Author Response · Author response to Decision Letter 0]

30 Nov 2022

Please see the response to reviewers document for details on the changes made.

---

## [Decision Letter · Decision Letter 1]

3 Mar 2023

PONE-D-22-21276R1Predicting fall risk using multiple mechanics-based metrics for a planar biped modelPLOS ONE

Dear Dr. Martin,

Thank you for submitting your manuscript to PLOS ONE. After careful consideration, we feel that it has merit but does not fully meet PLOS ONE’s publication criteria as it currently stands. Therefore, we invite you to submit a revised version of the manuscript that addresses the points raised during the review process.

We look forward to receiving your revised manuscript.

Kind regards,

Peter Andreas Federolf

Academic Editor

PLOS ONE

Journal Requirements:

Additional Editor Comments (if provided):

Reviewers' comments:

Reviewer's Responses to Questions

**Comments to the Author**

1. If the authors have adequately addressed your comments raised in a previous round of review and you feel that this manuscript is now acceptable for publication, you may indicate that here to bypass the “Comments to the Author” section, enter your conflict of interest statement in the “Confidential to Editor” section, and submit your "Accept" recommendation.

Reviewer #1: All comments have been addressed

Reviewer #2: All comments have been addressed

2. Is the manuscript technically sound, and do the data support the conclusions?

Reviewer #1: Yes

Reviewer #2: No

3. Has the statistical analysis been performed appropriately and rigorously? 

Reviewer #1: Yes

Reviewer #2: No

4. Have the authors made all data underlying the findings in their manuscript fully available?

Reviewer #1: Yes

Reviewer #2: Yes

5. Is the manuscript presented in an intelligible fashion and written in standard English?

Reviewer #1: Yes

Reviewer #2: No

6. Review Comments to the Author

Reviewer #1: The authors have improved the manuscript, and I feel it makes a nice contribution to the literature now. I have only one small comment regarding the new figure 2; please indicate (in the figure, and caption) what the 0.10 and 0.5 refer to

Reviewer #2: The paper is rejected as the authors did not cite the concerned paper and also the gives explanations are not convincing.

7. PLOS authors have the option to publish the peer review history of their article (what does this mean?). If published, this will include your full peer review and any attached files.

Reviewer #1: **Yes: **Sjoerd Bruijn

Reviewer #2: No

---

## [Author Response · Author response to Decision Letter 1]

8 Mar 2023

Please see the attached response document.

---

## [Editor Report · Decision Letter 2]

9 Mar 2023

Predicting fall risk using multiple mechanics-based metrics for a planar biped model

PONE-D-22-21276R2

Dear Dr. Martin,

We’re pleased to inform you that your manuscript has been judged scientifically suitable for publication and will be formally accepted for publication once it meets all outstanding technical requirements.

Kind regards,

Peter Andreas Federolf

Academic Editor

PLOS ONE

---

## [Editor Report · Acceptance letter]

15 Mar 2023

PONE-D-22-21276R2 

Predicting fall risk using multiple mechanics-based metrics for a planar biped model 

Dear Dr. Martin:

I'm pleased to inform you that your manuscript has been deemed suitable for publication in PLOS ONE. Congratulations! Your manuscript is now with our production department. 

Kind regards, 

on behalf of

Dr. Peter Andreas Federolf 

Academic Editor

PLOS ONE